# Pre-training Language Models for Comparative Reasoning

**Mengxia Yu[1], Zhihan Zhang[1], Wenhao Yu[1,2], Meng Jiang[1]**
[1]University of Notre Dame, [2]Tencent AI Seattle Lab
{myu2,zzhang23, mjiang2}@nd.edu, wenhaowyu@global.tencent.com

## Abstract

Comparative reasoning is a process of comparing objects, concepts, or entities to draw conclusions, which constitutes a fundamental cognitive ability. In this paper, we propose a novel framework to pre-train language models for enhancing their abilities of comparative reasoning over texts. While there have been approaches for NLP tasks that require comparative reasoning, they suffer from costly manual data labeling and limited generalizability to different tasks. Our approach introduces a novel method of collecting scalable data for text-based entity comparison, which leverages both structured and unstructured data. Moreover, we present a framework of pre-training language models via three novel objectives on comparative reasoning. Evaluation on downstream tasks including comparative question answering, question generation, and summarization shows that our pre-training framework significantly improves the comparative reasoning abilities of language models, especially under low-resource conditions. This work also releases the first integrated benchmark for comparative reasoning.

## 1 Introduction

Comparative reasoning constitutes a fundamental cognitive ability that plays a crucial role in decision-making. It refers to comparing objects, concepts, or entities to draw conclusions or make informed decisions. For example, consumers often compare products on their features such as price, quality, and user reviews before placing an order. Policy-makers weigh the advantages and disadvantages of different policy proposals to address pressing issues. Regarding textual documents, comparative reasoning is commonly needed in identifying differences between research studies, contrasting news articles from different sources, or synthesizing arguments of opposing viewpoints in a debate.

Recent research has developed models for a few NLP tasks related to comparing texts, including identifying comparative sentences (Jindal and Liu, 2006), mining comparable entities (Li et al., 2011), identifying comparative aspects from a set of questions (Bondarenko et al., 2022; Beloucif et al., 2022), extracting comparative summaries (Bista et al., 2019), and summarizing different opinions (Iso et al., 2022). Yet, the data collection for these tasks relies on expensive and time-consuming manual annotation. As a result, low-resource scenarios are common when it comes to new comparative tasks (Iso et al., 2022). Moreover, the task-specific design of such models limits their general comparative reasoning abilities. Meanwhile, pre-trained language models (PLMs) such as BART (Lewis et al., 2020) and T5 (Raffel et al., 2020) exhibit generalizability on several NLP tasks. However, existing pre-training methods such as masked language modeling and span in-filling fail to empower language models (LMs) with strong comparative reasoning abilities due to the lack of explicit training on comparisons.

To address these challenges, we propose a novel pre-training framework to enhance the comparative reasoning abilities of LMs. Specifically, it trains LMs to capture the comparison information between entities from paired documents. Our approach pilots around a scalable, labor-free data collection method that gathers documents as entity descriptions and a wealth of facts for entity comparison by combining structured (i.e., Wikidata) and unstructured data (i.e., news and Wikipedia). We represent these comparisons of facts as quintuples, which consist of a pair of entities and the corresponding values of their shared property. To empower LMs with comparative reasoning abilities on such data, given two comparable entities and their corresponding descriptive documents, we design three novel pre-training tasks including the generation of comparative answers, question-answer pairs, and summaries. Pre-training data of these tasks are obtained through automatic textualization of fac-

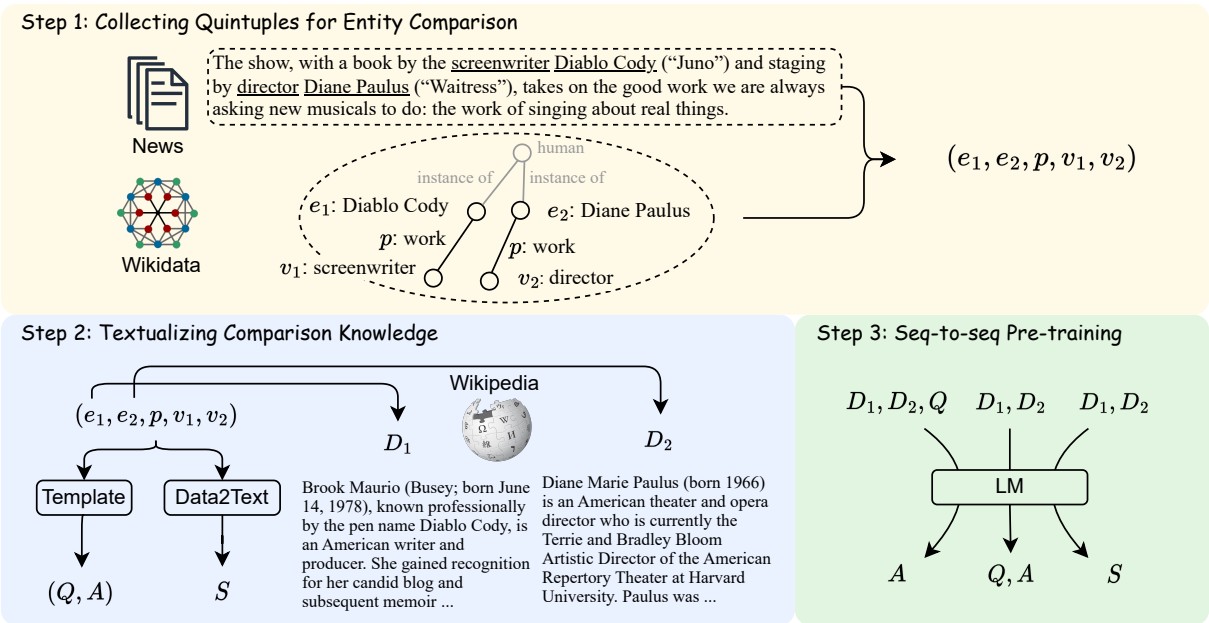

Figure 1: The framework of pre-training LMs for comparative reasoning abilities. In **Step 1**, we collect quintuples for entity comparison by combining structured knowledge base (i.e., Wikidata) and unstructured text corpora (i.e., Gigawords, CC-News, Wikipedia). Details are in § 3.2.2. In **Step 2**, to obtain text-based pre-training data, we textualize the quintuples into synthetic QA pairs with a set of templates, and convert the quintuples into summaries with an off-the-shelf data-to-text model. We gather Wikipedia documents as text descriptions of entities. Details are in §3.2.3. In **Step 3**, we design novel pre-training tasks for the LMs. Details are described in §3.3.

tual quintuples, so as to prevent expensive manual annotation. Subsequently, the pre-training tasks are uniformly formatted with natural language prompts to perform multi-task pre-training on the LMs. To our best knowledge, this work is the first to pre-train LMs for comparative reasoning.

To comprehensively evaluate the comparative reasoning abilities of LMs, we introduce a new benchmark with a suite of comparative reasoning tasks. It contains: (1) comparative question answering (QA), sourced from subsets of HotpotQA and 2WikiQA datasets (Yang et al., 2018; Ho et al., 2020); (2) comparative question generation (QG), including HotpotQG and 2WikiQG which are converted from the QA datasets; (3) comparative summarization, including the Diffen dataset that we crawled and the existing CocoTrip dataset (Iso et al., 2022).

With this benchmark, we conduct extensive experiments with vanilla PLMs (i.e., BART and T5) and their counterparts trained by our proposed framework. Results demonstrate a notable improvement in the performance of these PLMs on different comparative reasoning scenarios, especially under low-resource settings. Specifically, under the few-shot setting, the BART model pre-trained with our framework outperforms the vanilla BART by an average of 6.17 points on all datasets. Under the zero-

shot setting, the improvement becomes as high as 13.99 points on average. These results highlight the effectiveness of our pre-training framework which empowers LMs with impressive abilities of comparative reasoning even when zero or few examples are available. Besides, we analyze the effect of the pre-training data size on the model performance, and provide a case study to better understand the benefits of our pre-training.

Our contributions are summarized as follows:

- We propose a scalable method of collecting and designing training data for entity comparison, using both structured and unstructured data sources that are publicly accessible.
- We present a novel framework for pre-training LMs to enhance their comparative reasoning abilities on multiple related objectives.
- We provide the first benchmark for entity comparison over texts, serving as a foundation for future research in this topic.

## 2 Related Work

### 2.1 Comparative Reasoning

The academic landscape of comparative reasoning tasks has seen a significant progression. Early research primarily focused on mining explicit comparative information from massive corpora, such

as identifying comparative sentences (Jindal and Liu, 2006), extracting comparable entities (Li et al., 2011), and classifying components of comparison (Beloucif et al., 2022). Recent work focused more on text generation tasks such as generating arguments to answer comparative questions (Chekalina et al., 2021), generating comparable questions from news (Beloucif et al., 2022), and summarizing comparative opinions (Lerman and McDonald, 2009; Iso et al., 2022). The existing techniques were designed for specific tasks and could not generalize across all types of comparative reasoning tasks. Moreover, they suffered from the scarcity of labelled data in low-resource settings. Our approach aims to address these two challenges.

## 2.2 Language Model Pre-training

It is worth exploring to use both structured and unstructured data in language model pre-training. Early work proposed to fuse knowledge graphs and textual information by encoding entities or nodes as a part of the input (Zhang et al., 2019; Wang et al., 2021; Yu et al., 2022; Ke et al., 2021; Liu et al., 2021; Hu et al., 2022). For example, Hu et al. (2022) integrated graph-based knowledge augmented modules to bring structured knowledge into generative LMs. Another branch of work incorporated entity information (Xiong et al., 2020; Zhang et al., 2022) or relational information (Qin et al., 2021; Hu et al., 2021) without modifying the structure of the LM. While these pre-trained models delivered encouraging outcomes across a multitude of downstream tasks, they were not tailored for the needs of comparative reasoning. A novel design of pre-training objectives is necessary, which has not been inherently presented in these models. Regarding the collection of pre-training data, RGPT-QA (Hu et al., 2021) combined Wikidata and Wikipedia to generate synthetic QA pairs for pre-training. However, such a set of pre-training data only comprised statements of individual entities, so the trained model is not effective for multi-hop comparative questions. MQA-QG (Pan et al., 2021) and MuSiQue (Trivedi et al., 2022) generate synthetic data for unsupervised multi-hop QA, but they did not consider other comparative tasks.

## 3 Pre-training Framework

To enhance the ability of LMs in comparative reasoning, we introduce a novel framework for pre-training LMs on a collected corpus of comparative entities. Specifically, LMs are given a pair of documents, each describing an entity, and are trained to generate target sequences which require comparison between the entities. We consider three types of target sequences: an answer to a comparative question, a question-answer pair that requires comparative reasoning, and a comparative summary of entities. Correspondingly, we design three text-to-text pre-training tasks that require the LMs to simultaneously attend to both documents and extract information for pairwise comparison. This framework enables them to handle various downstream scenarios that require comparative reasoning.

To collect data for large-scale pre-training, we extract comparable entity pairs with their properties by combining structured and unstructured data. The extracted data are first formulated as quintuples (elaborated in §3.1 to show their comparative nature), which are later used for text-to-text pre-training.

## 3.1 Notations

A Wikidata statement is denoted as $(e, p, v)$. Here, $e$ signifies the entity, which is the subject of the statement. $p$ refers to the property, describing the aspect of the entity that the statement addresses. $v$ represents the value, which is the object entity or specific value associated with the property. We define a quintuple as a pair of Wikidata statements of two comparable entities on a common property. Formally, a **quintuple** is represented as $(e_1, e_2, p, v_1, v_2)$, where $p$ is a common property of $e_1$ and $e_2$, and $v_1$ and $v_2$ are the corresponding values. Such quintuples enable the comparison on shared properties, reflecting the similarity or difference between the corresponding property values or tail entities.

In our framework, the input sequence constitutes two documents $D_1$ and $D_2$ on $e_1$ and $e_2$ respectively. The target sequences are textualized forms of the quintuple, such as question-answer pairs (denoted by $(Q, A)$), and summaries (denoted by $S$).

## 3.2 Pre-training Data Preparation

### 3.2.1 Data Sources

Structured data is a reliable source for obtaining entity information. We use Wikidata, a collaborative knowledge base that stores data in a structured format. Wikidata contains numerous statements that describe entities, where each statement includes a property of the entity and a value. Each entity and

**All:**
Q: Do $e_1$ and $e_2$ have the same/different value of $p$?
A: Yes/No
Q: Do $e_1$ and $e_2$ both have the value of $v_1$ in terms of $p$?
A: Yes/No
Q: What are the $p$ of $e_1$ and $e_2$?
A: $v_1$, $v_2$

**If $v_1 \neq v_2$:**
Q: Which one of the following entity's $p$ is $v_1$? $e_1$ or $e_2$?
A: $e_1$
Q: Is $e_1$'s $p$ $v_1$ or $v_2$?
A: $v_1$

**If $v_1 = v_2$:**
Q: Which entity has the same value as $e_1$ in terms of $p$?
A: $e_2$
Q: $e_1$ and $e_2$ are known for what (value) of $p$?
A: $v_1/v_2$

Table 1: Synthetic QA templates. **All** indicates the templates are applied to all quintuples. The templates under **If $v_1 \neq v_2$:** or **If $v_1 = v_2$:** are applied to quintuples whose $v_1$ and $v_2$ are different or the same, respectively.

property is associated with a set of aliases.

Unstructured data including news sources (i.e., Gigawords, CC-News) and encyclopedia (i.e., Wikipeda) offer an abundance of information for determining the comparability of entities and relevant properties. For example, a sentence in a piece of news from New York Times like *"The show, with a book by the screenwriter* Diablo Cody *('Juno') and staging by director* Diane Paulus *('Waitress'), takes on the good work ...,"* indicates that Diablo Cody and Diane Paulus can be compared on the property of *work* (values: *screenwriter* vs. *director*). Besides, Wikipedia contains a vast collection of articles pertaining to a large set of entities. A Wikidata entity uniquely corresponds to a Wikipedia article whose title matches the entity's surface form.

### 3.2.2 Quintuple Collection

In this section, we elaborate the process of collecting quintuples by combining structured data (i.e., Wikidata) and unstructured data (i.e., news and Wikipedia). Intuitively, when a pair of statements concerning the same property of related entities co-occur in a textual context, there is a high probability that these statements are indeed comparable.

To extract this comparability information, we first sample a paragraph from the news or Wikipedia. Then, we link Wikidata statements to the sentences in the paragraph by identifying the mentions of entity $e$, property $p$, and value $v$ using string matching. Specifically, a statement $(e, p, v)$

is linked to a sentence if the aliases of $e$, $p$, and $v$ all appear in the sentence. Next, we pair $(e_1, p_1, v_1)$ and $(e_2, p_2, v_2)$ if they satisfy the following criteria:

1. $e_1$ and $e_2$ belong to the same category, e.g., both have the value *human* for property *instance of*. This ensures the entities are analogous to each other.
2. $p_1 = p_2$. This follows the common practice that comparisons are usually made on a shared property between two entities.
3. The sentences linked to $(e_1, p_1, v_1)$ and $(e_2, p_2, v_2)$ co-occur in a from news or Wikipedia. Being mentioned together indicates implicit entity comparison.

We denote such a statement pair as a quintuple $(e_1, e_2, p, v_1, v_2)$. By following the above criteria, such quintuples store necessary information for comparing entities $e_1$ and $e_2$, which plays a critical role in our pre-training task design.

### 3.2.3 Quintuple Textualization

In order to empower the LM with the ability of comparative reasoning in various language generation scenarios, we pre-train the LM in a text-to-text manner. To achieve this, the first step is to represent the comparative information inherent in the quintuples in a textual form.

To begin with, we extract descriptive documents $D_1$ and $D_2$ for each pair of entities $e_1$ and $e_2$ as contexts in our pre-training. First, we find Wikipedia articles of $e_1$ and $e_2$ by the links from Wikidata. To ensure the information within the quintuple can be inferred from the context, we filter the articles based on whether any sentence within the article can be linked to statements $(e_1, p, v_1)$ and $(e_2, p, v_2)$. We link the statements based on two heuristics: (1) Within an article pertaining to entity $e$, sentences are highly probable to discuss $e$ as their subject; (2) If a sentence in a Wikipedia article of $e$ mentions both $e$ and $v$ from a Wikidata statement $(e, p, v)$, then the sentence is likely to describe the fact of $(e, p, v)$. Thus, we link the statements to sentences whenever $(e, v)$ or $(p, v)$ can be matched. To assess the linking quality, we randomly sampled 100 statement-sentence links and performed manual inspection. The linking accuracy exceeds 95%, indicating the Wikidata statements are effectively linked to the sentences. Finally, due to length limit of LMs, we split the original article into 10-sentence segments, and use the segment that contains the linked sentence as the

| Pre-training Task | Source → Target |
|---|---|
| Comparative Answer Generation | Answer the comparative question. Question: $\{Q\}$ Context: $\{D_1\}$ [SEP] $\{D_2\} \rightarrow A$ |
| Comparative QA Pairs Generation | Generate a comparative question-answer pair. Context: $\{D_1\}$ [SEP] $\{D_2\} \rightarrow Q; A$ |
| Comparative Summary Generation | Generate a comparative summary. Context: $\{D_1\}$ [SEP] $\{D_2\} \rightarrow S$ |
| Text Infilling | $\{$corrupted $D_1\}$ [SEP] $\{$corrupted $D_2\} \rightarrow \{D_1\}$ [SEP] $\{D_2\}$ |

Table 2: Task names and the format of source-target sequence format in each pre-training task.

---

**Quintuple**: (John Lewis, Hank Johnson, member of political party, Democratic Party, Democratic Party)

---

**D$_2$**: John Robert Lewis (February 21, 1940July 17, 2020) was an American statesman and civil rights activist who served in the United States House of Representatives for from 1987 until his death in 2020. He was the chairman of the Student Nonviolent Coordinating Committee (SNCC) from 1963 to 1966. ... While in the House, Lewis was one of the leaders of the Democratic Party, serving from 1991 ...

---

**D$_2$** Henry Calvin Johnson Jr. (born October 2, 1954) is an American lawyer and politician serving as the U.S. representative for since 2007. He is a member of the Democratic Party. ...

---

**Synthetic QA pairs Q, A:**
1. Q: Do John Lewis and Hank Johnson have the same value for member of political party? A: Yes
2. Q: Do John Lewis and Hank Johnson both have the value of Democratic Party in terms of member of political party? A: Yes
3. Q: What are the member of political party of John Lewis and Hank Johnson? A: Democratic Party
4. Q: Which entity has the same value as John Lewis in terms of member of political party? A: Hank Johnson
5. Q: John Lewis and Hank Johnson are known for what value of member of political party? A: Democratic Party

---

**Synthetic Summary S**: John Lewis is a member of the Democratic Party, as is Hank Johnson.

---

Table 3: A quintuple for comparison between John Lewis and Hank Johnson on their shared property member of political party. The example consist of the textualized data used in pre-training, including the entities' descriptive documents $D_1$ and $D_2$, QA pairs $(Q, A)$ synthesized with designed templates, and the synthetic summary $S$ generated by a data-to-text model with two the two Wikidata statements as input.

descriptive document $D_1$ for $e_1$ (or $D_2$ for $e_2$).

Next, we convert the comparison knowledge encapsulated within the quintuples into comparative texts, namely, QA pairs and summaries. To synthesize comparative QA pairs $(Q, A)$, we design a diverse set of templates shown in Table 1. To generate synthetic comparative summaries $S$, we utilize an off-the-shelf data-to-text model (Ribeiro et al., 2021) fine-tuned on DART (Nan et al., 2021) dataset. This allows us to transform quintuples into concise declarative sentences. An example of a textualized quintuple is provided in Table 3.

## 3.3 Pre-training Tasks and Objectives

In this section, we describe three comparative pre-training tasks used to train LMs. They are all text generation tasks, which align seamlessly with architectures of widely used language models such as BART and T5. We unify them with task-specific prompts in a multi-task setting, shown in Table 2.

### 3.3.1 Comparative Answer Generation

To train the LM with the ability to answer comparative questions, given a comparative question, we concatenate it with the documents $D_1, D_2$ as input, and then train the model to generate the corresponding answer. This task not only requires the model to find relevant contexts to the question in each single document, more importantly, it requires the interaction between both documents to make the comparison. We define the loss function as:

$$\mathcal{L}_{\text{QA}} = - \sum_{(Q_i, A_i) \in \mathcal{T}} \log P(A_i | Q_i, D_1, D_2)$$

in which $\mathcal{T}$ is a set of QA pairs derived from the templates. and $P(\cdot)$ is the predicted probability.

### 3.3.2 Comparative QA Pairs Generation

Given two documents, the model is required to generate comparative questions and answers. With this objective, the model learns to attend to both documents, identify the comparable properties of two entities and ask meaningful questions:

$$\mathcal{L}_{\text{QAG}} = - \sum_{(Q_i, A_i) \in \mathcal{T}} \log P(Q_i, A_i | D_1, D_2)$$

### 3.3.3 Comparative Summary Generation

Comparative summarization aims at generating summaries that highlight the similarities or differences between two entities given their descriptions. Given two documents, the model is tasked with generating short comparative summaries that represent the comparable statements:

$$\mathcal{L}_{\text{SUM}} = - \sum_{S \in \mathcal{S}} \log P(S | D_1, D_2)$$

where $\mathcal{S}$ is the set of summaries from quintuple textualization.

### 3.3.4 Prompt-based Multi-task Training

Inspired by the prompt-based multi-task training methods utilized by previous text-to-text transformers (Raffel et al., 2020; Sanh et al., 2022), we jointly train the aforementioned pre-training tasks by unifying their input sequences with natural language prompts. The detailed format of source and target sequences are shown in Table 2. The model is jointly optimized for all tasks, which encourages the model to learn generalizable representations that are beneficial across tasks. To preserve its general language modeling ability, we employ the proposed pre-training tasks along with the text infilling (TI) task, where the model is required to reconstruct the documents corrupted with randomly masked spans, as described in Lewis et al. (2020). We denote the loss function for text infilling as as $\mathcal{L}_{\text{TI}}$. Hence, the overall objective is as follows: $\mathcal{L} = \mathcal{L}_{\text{QA}} + \mathcal{L}_{\text{QAG}} + \mathcal{L}_{\text{SUM}} + \mathcal{L}_{\text{TI}}$.

We denote the proposed multi-task pre-training for comparison as +CMP. To analyze the effects of each pre-training task, we define single-task variants: +CMP$_{\text{QA}}$ for comparative answer generation, +CMP$_{\text{QAG}}$ for comparative QA pairs generation, and +CMP$_{\text{SUM}}$ for summary generation.

## 4 Experiments

### 4.1 Datasets and Evaluation Metrics

To evaluate our proposed method, we consider downstream tasks involving comparative reasoning, including comparative question answering (QA), comparative question generation (QG) and comparative summarization. In this section, we introduce the downstream datasets and evaluation metrics.

### 4.1.1 Comparative Question Answering

Comparative QA requires the comparison of two entities on their shared properties. Since our focus on comparison over documents instead of knowledge retrieval, we do not include distractor passages but directly use the gold evidence passages as the context for question answering. For evaluation, we calculate the exact match (EM) score between the predicted answer and the ground-truth answer, after necessary normalization (Chen et al., 2017). Besides, unigram F-1 scores are also calculated as a complementary metric.

**HotpotQA and 2WikiQA.** HotpotQA (Yang et al., 2018) and 2WikiMultihopQA (2WikiQA) (Ho et al., 2020) are factual question answering datasets collected from English Wikipedia. These datasets require multi-hop reasoning on different entities before reaching the correct answer. To focus on comparative ability, we obtain the subset of comparative questions by their question type annotations in the original dataset. As a result, the train and validation set of HotpotQA consist of 17,456 and 1,487 instances, respectively. Likewise, 2WikiQA comprises 51,693 and 3,040 instances in training and validation set, respectively. We report results in validation sets.

### 4.1.2 Comparative Question Generation

Comparative QG aims at generating questions that draw comparisons between the shared properties of two entities, given their textual descriptions. We convert the aforementioned QA datasets to QG by using the evidence passages as input and the comparative question as output. We report the results on validation sets using overall BLEU (Papineni et al., 2002) and ROUGE-L (Lin, 2004) metrics.

### 4.1.3 Comparative Summarization

Comparative summarization aims at generating summaries that highlight the similarities or differences between two entities given their descriptions. Following the convention in text summarization (Zhang et al., 2020), we evaluate the generated summaries with ROUGE-2 and ROUGE-L scores.

**CocoTrip.** We collect data from the common opinion summarization setting of the CocoTrip dataset (Iso et al., 2022), which involves summarizing the shared opinions from two sets of reviews about two hotels. The dataset consists of 20, 10, and 18 instances for training, validation and test set, respectively. Since the test data is available, we report the results on the test set. We concatenate both reviews as the input context.

**Diffen.** To address the lack of available datasets for the comparative summarization of two entities, we create a new dataset from Diffen.com, a website recognized for offering high-quality, human-authored comparisons between different people or objects to help people make informed decisions. Comparison articles on Diffen.com typically include a brief introduction summarizing the similarities and differences. We manually collect these introductory paragraphs as comparative summaries. To gather input sources, we obtain Wikipedia articles for each entity. The resulting dataset comprises 20 instances for training and 100 instances for vali-

| | Comparative QA | | | | Comparative QG | | | | Comparative Summarization | | | | |
| | HotpotQA | | 2WikiQA | | HotpotQG | | 2WikiQG | | CocoTrip | | Diffen | | |
| | EM | F1 | EM | F1 | BLEU | R-L | BLEU | R-L | R-2 | R-L | R-2 | R-L | AVG |
|---|---|---|---|---|---|---|---|---|---|---|---|---|---|
| ChatGPT | 73.45 | 62.68 | 82.95 | 74.33 | 7.36 | 29.16 | 10.89 | 33.02 | 6.80 | 18.59 | 9.29 | 21.69 | 35.85 |
| **Full-data** BART | **69.27** | **75.70** | **91.87** | **92.43** | 16.29 | 43.41 | 35.28 | 61.94 | 23.63 | 44.99 | 10.04 | 24.39 | 49.10 |
| + CMP | 69.26 | 75.43 | 91.81 | 92.30 | **17.18** | **43.66** | **35.82** | **62.13** | **27.60** | **47.90** | **12.11** | **26.69** | **50.16** |
| T5 | **73.16** | **79.20** | 87.40 | 89.67 | **17.57** | **44.70** | 36.02 | 62.70 | 29.18 | 45.57 | **9.21** | **24.03** | 49.87 |
| + CMP | 72.69 | 78.83 | **88.75** | **91.08** | 17.26 | 44.65 | **36.12** | **63.18** | **30.48** | **49.19** | 8.12 | 23.04 | **50.28** |
| **Few-shot** BART | 33.82 | 39.70 | 37.65 | 39.67 | 11.38 | 39.04 | 30.02 | **57.14** | 23.63 | 44.99 | 10.04 | 24.39 | 32.62 |
| + CMP | **44.31** | **52.15** | **57.58** | **58.49** | **12.75** | **39.33** | **30.29** | 56.28 | **27.60** | **47.90** | **12.11** | **26.69** | **39.09** |
| T5 | 48.89 | 54.71 | 43.85 | 45.63 | 6.48 | 30.95 | 6.71 | 28.44 | 29.18 | 45.57 | **9.21** | **24.03** | 31.14 |
| + CMP | **50.50** | **58.29** | **56.51** | **58.33** | **8.18** | **33.88** | **12.12** | **37.95** | **30.48** | **49.19** | 8.12 | 23.04 | **35.55** |
| **Zero-shot** BART | 0.00 | 11.93 | 0.00 | 19.30 | 1.70 | 18.53 | 3.45 | 20.34 | 4.09 | 18.32 | 6.32 | 17.90 | 10.16 |
| + CMP | **31.47** | **39.04** | **40.55** | **42.47** | **6.86** | **29.13** | **9.21** | **32.23** | **6.11** | **24.43** | **8.02** | **20.22** | **24.15** |
| T5 | 20.44 | 28.87 | 20.88 | 26.92 | 1.21 | 18.70 | 2.38 | 18.53 | 8.94 | 25.28 | 5.61 | 17.65 | 16.28 |
| + CMP | **44.25** | **52.62** | **54.34** | **56.30** | **7.24** | **28.99** | **5.83** | **32.31** | 8.63 | **28.19** | **5.72** | **18.43** | **28.57** |

Table 4: Main results. Our pre-trained models denoted by +CMP, bring *significant* performance gain to BART and T5 in zero-shot (e.g., relatively +82% and +220% of F1 on HotpotQA) and few-shot (e.g., relatively +29% and +52% of F1 on 2WikiQA) settings across all tasks. In full-data settings that assume a huge number of labeled examples are available, our approach makes smaller improvements on the two models.

dation. The task aims at generating a comparative summary based on the given text descriptions of two entities. The input sequence consists of concatenated entity descriptions, with each description truncated to the first 512 tokens.

## 4.2 Experimental Setup

As a pilot study on pre-training for comparative reasoning, we adopt the pre-trained BART (Lewis et al., 2020) and T5 (Raffel et al., 2020) as baselines. Models further pre-trained on our comparative objectives are denoted as BART+CMP and T5+CMP, respectively. See training details in A.2. We also conduct zero-shot experiments with Chat-GPT (gpt3.5-turbo), where the details are in 3.2.3. Since ChatGPT is pre-trained on much larger-scale data, and the downstream datasets might have leaked to its training data, it is not a comparable baseline. We provide ChatGPT (OpenAI, 2021) as a reference to the performance of one of the most advanced large language models.

To test the comparative reasoning ability of models under low-resource scenarios, we compare the models in few-shot and zero-shot settings in addition to the conventional full-data fine-tuning. In the few-shot setting, we randomly selected 100 instances from the training set. However, given the limited number of training instances available in CocoTrip and Diffen (only 20 instances each), we

merge the full-data and few-shot settings for these two datasets. See training details in Appendix A.3.

## 4.3 Experimental Results

### 4.3.1 Effects of Comparative Pre-training

In the comprehensive evaluation across the aforementioned six datasets, we compare LMs trained with our method against the vanilla BART and T5. Main results are listed in Table 4.

When abundant data are available, both our proposed models, BART+CMP and T5+CMP, achieve competitive performance which improves their corresponding baselines by ~1 point on average. However, in low-resource scenarios, the superiority of our method over the baselines becomes clearly evident. Specifically, for the few-shot setting, our BART+CMP achieves an average score of 39.09, showing an relative improvement of +19.8% compared to BART's score of 32.62. Similarly, our T5+CMP achieves an average score of 35.55, which improves +14.2% relatively over T5. Among three tasks, our models show the most significant improvement on comparative QA, demonstrating the effectiveness of our synthetic QA pre-training. In zero-shot setting, BART+CMP and T5+CMP also consistently surpass their baselines by large margins. For instance, BART+CMP achieves an average score of 24.15, which outperforms BART by +13.99 (+137% relatively). Likewise, T5+CMP

| | | Comparative QA | | | | Comparative QG | | | | Comparative Summarization | | | | |
| | | HotpotQA | | 2WikiQA | | HotpotQA | | 2WikiQA | | CocoTrip | | Diffen | | |
| | | EM | F1 | EM | F1 | BLEU | R-L | BLEU | R-L | R-2 | R-L | R-2 | R-L | AVG |
|---|---|---|---|---|---|---|---|---|---|---|---|---|---|---|
| Few-shot | BART | 33.82 | 39.70 | 37.65 | 39.67 | 11.38 | 39.04 | 30.02 | 57.14 | 23.63 | 44.99 | 10.04 | 24.39 | 32.62 |
| | + CMP | 44.31 | 52.15 | **57.58** | **58.49** | **12.75** | 39.33 | 30.29 | 56.28 | 27.60 | 47.90 | 12.11 | 26.69 | **39.09** |
| | + CMP$_{QA}$ | **45.25** | **52.46** | 55.96 | 57.22 | 11.76 | **39.51** | 25.73 | 52.43 | 24.91 | 45.36 | **12.36** | 25.92 | 37.41 |
| | + CMP$_{QAG}$ | 32.21 | 38.17 | 45.13 | 46.63 | 12.57 | 39.29 | 26.89 | 52.74 | 27.22 | 42.55 | 12.28 | **26.85** | 33.54 |
| | + CMP$_{SUM}$ | 32.41 | 37.81 | 32.34 | 34.33 | 12.50 | 39.40 | **33.04** | **59.29** | **30.54** | 47.84 | 12.10 | 26.69 | 33.19 |
| Zero-shot | BART | 0.00 | 11.93 | 0.00 | 19.30 | 1.70 | 18.53 | 3.45 | 20.34 | 4.09 | 18.32 | 6.32 | 17.90 | 10.16 |
| | + CMP | 31.47 | 39.04 | 40.55 | 42.47 | **6.86** | **29.13** | **9.21** | 32.23 | **6.11** | 24.43 | 8.02 | **20.22** | **24.15** |
| | + CMP$_{QA}$ | **34.50** | **42.41** | **43.85** | **45.83** | 1.57 | 19.39 | 3.31 | 20.27 | 0.00 | 5.14 | 1.61 | 6.72 | 18.71 |
| | + CMP$_{QAG}$ | 0.00 | 13.79 | 0.00 | 20.85 | 6.16 | 27.89 | 8.00 | 29.44 | 1.39 | 16.62 | 6.13 | 18.22 | 12.37 |
| | + CMP$_{SUM}$ | 0.00 | 16.56 | 0.00 | 20.02 | 5.33 | 32.78 | 6.76 | **32.78** | 5.53 | **25.79** | **8.34** | 20.00 | 14.40 |

Table 5: Few-shot and Zero-shot results of models with multi-task pre-training (denoted by +CMP) vs. single-task pre-training (denoted by +CMP$_{QA}$, +CMP$_{QAG}$, and +CMP$_{SUM}$).

achieves an average score of 28.57, improving +75% over T5 relatively. These results indicate that our proposed pre-training method greatly enhances LM's performance in low resource scenarios, while retaining competitive performance when abundant training data are available.

### 4.3.2 Effects of Pre-training Tasks

To further explore the benefits of multi-task pre-training, we compare the performance of our models pre-trained on any single task (i.e., QA, QAG or SUM) with the unified models pre-trained on all proposed tasks. Results are shown in Table 5. When the model is pre-trained on a single task, we observe a significant improvement in performance on the downstream task that closely resembled the pre-training task. However, such models do not exhibit similar improvements on other tasks that are less similar in nature. For example, BART+CMP$_{QA}$ improves over BART by a large margin on few-shot comparative QA (+11.43 points in F1 on HotpotQA), but performs at a similar or lower level as BART on QG (+0.38 points in BLEU on HotpotQG and -4.29 points in BLEU on 2WikiQG). On the other hand, the unified model BART+CMP exhibits substantial improvements across all downstream tasks and therefore achieves the best overall performance. The improvements brought by multi-task pre-trained on each task is comparable to the gains achieved through the corresponding task-specific pre-training. These results suggest that pre-training on a single task enhances the model's ability to transfer knowledge only to tasks with similar characteristics, while multi-task pre-training enables the model to learn more gen-

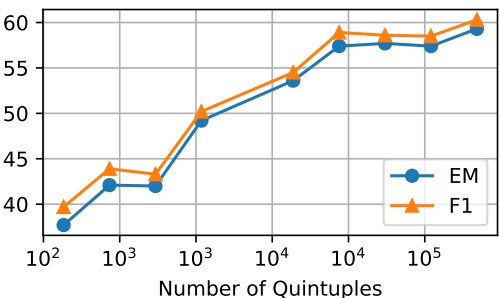

Figure 2: Few-shot performance (measured by F1) of BART+CMP on 2WikiQA, when the model is pre-trained on different number of quintuples.

eralized representations and to effectively transfer the shared knowledge across different tasks.

### 4.4 Effects of the Size of Pre-training Data

In Figure 2, we plot the few-shot performance of BART+CMP on 2WikiQA according to the number of quintuples used in pre-training. We observe that when the number of quintuples increases on a logarithmic scale, the performance grows linearly. The analysis reveals that scaling the pre-training data benefits the downstream tasks, affirming the effectiveness of the proposed method for gathering large scale pre-training data. Further discussion on the effects of entity coverage is shown in Appendix 4.4.1.

### 4.4.1 Effects of Entity Coverage

To study the effects of the downstream entity coverage in pre-training data, we count the overlapping entities between pre-training quintuples and HotpotQA validation set. We proceed by removing the overlapping entities from pre-training data, fol-

lowed by pre-training the model and evaluating on the same HotpotQA validation set. Our findings, as depicted in Table 6, demonstrate that even with the removal of overlapping entities, where 97.6% of the pre-training data persists, the model's few-shot performance remains strong. Hence, it can be concluded that the observed performance improvement is not attributed to downstream entity coverage, instead, it is attributed to the task adaptation, indicating the potential generalizability to unseen entities.

| # Quintuples | Covered Ent. | EM | F1 |
|---|---|---|---|
| 219,598 | 471 (18%) | 44.92 | 52.41 |
| 214,327 | 0 | 44.86 | 52.38 |

Table 6: Few-shot performance of HotpotQA. The quintuples in our overall pre-training cover 18% of the downstream entities. When downstream entities are excluded from pre-training, a substantial amount (97.6%) of pre-training data remains, and the downstream performance remains robust.

## 4.5 Case Study

To intuitively show the comparative reasoning ability of our pre-trained model, we present an example of comparative summarization in Table 7. Given documents describing airsoft and paintball, models are expected to generate a summary comparing the commonalities and differences of these two games. However, without exhaustive fine-tuning, the generated summary of BART fails to describe the correct relationship between these two entities. On the contrary, after pre-trained on various comparative reasoning objectives, our model generates high quality comparative summaries based on the provided documents under the few-shot setting. The generated summary includes that both games are popular shooting sports while also comparing their differences in their equipment.

## 5 Conclusion

In this paper, we presented a novel framework for pre-training language models for comparative reasoning. It obtained quintuples for entity comparison by combining structured and unstructured data, converted the quintuples into textual components, and employed them in three novel sequence-to-sequence pre-training tasks. We demonstrated the effects of the pre-training tasks on six downstream datasets, especially in limited-resource scenarios. To facilitate the assessment of models' capability

**$D_1$**: Airsoft is a team game in which participants eliminate opposing players by tagging them out of play with spherical plastic projectiles shot with mock air weapons called airsoft guns. ... (446 words left)
**$D_2$**: Paintball is a competitive team shooting sport in which players eliminate opponents from play by hitting them with spherical dye-filled gelatin capsules called paintballs that break upon impact. ... (472 words left)

**Gold**: Airsoft is a popular combat simulation game where participants are eliminated when hit by pellets launched from guns that resemble real firearms. In paintball participants try to hit each other with paintballs launched from a special paintball marker/gun. While airsoft is cheaper and provides a more realistic warfare experience, paintball is more popular, more organized and has larger events.

**BART** (R-L: 18.66, R-2: 4.39)
Airsoft is a team shooting sport in which participants eliminate opponents by hitting them with airsoft guns. Airsoft guns are shaped like basketballs or baseball bats and are equipped with a series of round-shaped projectiles called paintballs.
**BART+CMP** (R-L: 19.17, R-2: 8.62)
Airsoft and Paintball are two of the most popular shooting sports of all time. Airsoft is a shooting sport that involves hitting opponents with airsoft guns, while Paintball is a more aggressive game that uses a softer, more aggressive, ball-shaped paintball.

Table 7: A test example of Diffen dataset. BART and BART+CMP refer to the model predictions under few-shot fine-tuning. BART+CMP generated the similarities and differences between airsoft and paintball.

of entity comparison over texts, we release a benchmark for future research.

## Limitations

In our pre-training framework, we generate synthetic data with templates to collect comparative question-answer pairs, which may cause fluency issues on some synthetic questions. Such noise in the pre-training data might affect the downstream performance. Similarly, since the synthetic summaries were generated by an off-the-shelf data-to-text model, the language of generated summaries can be rigid and lack of diversity and flexibility. Future work can adopt more advanced approaches to convert quintuples into more fluent and diverse texts for pre-training. Another limitation is that BART and T5 have a maximum input token limit of 1,024. When dealing with longer documents or complex comparative scenarios, this limation may lead to truncation of relevant context, potentially affecting the model's performance. Future work can explore LMs that can handle longer texts.

## Acknowledgements

This work was supported by NSF IIS-2119531, IIS-2137396, IIS-2142827, IIS-2234058, CCF-1901059, and ONR N00014-22-1-2507. Wenhao Yu is also supported by Bloomberg Data Science Ph.D Fellowship.

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

## A  Experiment Details

| | |
|---|---|
| Batch size | 192 |
| Learning rate | 3e-5 |
| Warmup ratio | 1% |
| Max source length | 512 |
| Max target length | 512 |
| Training steps | 70k |

Table 8: Pre-training hyperparameters.

### A.1  Pre-training Data

For Wikidata, we use the dump wikidata-20220103-all. For Gigawords we use the dump from `https://catalog.ldc.upenn.edu/LDC2011T07`. For CC-News we download the data from `https://huggingface.co/datasets/cc_news`. We randomly sample a small portion of data as validation set. An example of pre-training data is shown in Table 3.

| | Train | Validation |
|---|---|---|
| All | 219,598 | 6,252 |
| Gigawords | 110,399 | 3,179 |
| Wikipedia | 90,420 | 2,557 |
| CC-News | 18,779 | 516 |

Table 9: Number of quintuples in pre-training data from each unstructured data sources.

### A.2  Pre-training Details

Our models are initialized from checkpoints `facebook/bart-base` and `t5-base` and further trained with our pre-training tasks. All models are implemented with Hugging Face Transformers 4.17. The hyperparameters used in pre-training are listed in Table 8. For text infilling, we mask 30% of the tokens. The pre-trained model checkpoints are selected by the lowest validation loss.

### A.3  Downstream Experimental Details

For downstream experiments, we fine-tune the models with a batch size of 64, and search for learning rates among 1e-5, 3e-5, 1e-4. For QA and QG, we set the max input length as 512 tokens and max output length as 32. For summarization, we set the max input length as 1,024 tokens and the output length as 128 tokens. For comparative QA, we select the best checkpoints by the highest F1. For comparative QG, we select the best checkpoints by the highest BLEU. For comparative summarization, the best checkpoints are selected by the highest ROUGE-L. For evaluation metrics, we adopt the implementations of Exact Match, unigram F1, and ROUGE-L by KILT (Petroni et al., 2021), and the BLEU implemented by the Hugging Face evaluate library (v0.3.4). For all downstream datasets, we report the average scores of three run with different random seeds.

### A.4  Experimental Details of ChatGPT

Since ChatGPT is a much larger model with much more pre-trained data, and the downstream datasets might have leaked to its training data, we provide the result of ChatGPT only as a reference to the performance of one of the most advanced large language models. We prompt ChatGPT (gpt3.5-turbo) in zero-shot setting and the prompts are shown in the below table. We modify the prompts from the ones we use with T5/BART and empirically choose the ones that work effectively, as shown in Table 10.

| Task | Prompt |
|---|---|
| QA (HotpotQA, 2WikiQA) | Paragraph: $[D_1, D_2]$ 
 Question: $[Q]$ 
 Only output the short answer. |
| QG (HotpotQG, 2WikiQG) | Paragraph: $[D_1, D_2]$ 
 Given the paragraphs above, please output a factoid question that requires comparison of the two entities on some shared property. |
| Summarization (CocoTrip) | Reviews of Hotel 1: $[D_1]$ 
 Reviews of Hotel 2: $[D_2]$ 
 Given the reviews of the two hotels above, please output a summary of the common opinions about both hotels, which only contains subjective information that is commonly described in both sets of the reviews. The summary should be less than 30 words. |
| Comparative Summ (Diffen) | Paragraph 1: $[D_1]$ 
 Paragraph 2: $[D_2]$ 
 Based on the paragraphs above, output a comparative summary of the two entities. The summary should be less than 100 words. |

Table 10: Prompts of comparative downstream tasks for ChatGPT.