# OpenReview forum: "Pre-training Language Models for Comparative Reasoning"
_EMNLP/2023/Conference — EMNLP 2023 Main_

### Official Review · Reviewer_W4mp · 2023-07-31

**Soundness:** 3

**Excitement:**

4: Strong: This paper deepens the understanding of some phenomenon or lowers the barriers to an existing research direction.

**Missing References:**

None I can think of.

**Paper Topic And Main Contributions:**

This paper focuses on and builds a foundation for research in comparative reasoning, a problem with limited resources. The authors make three main contributions:
1. A scalable data collection method for comparative reasoning that mines entities from structured resources such that the entities have common properties over which they can be compared, then automatically retrieves text documents describing these entities and their properties through a heuristic approach.
2. A new pre-training paradigm for LMs that enhances their ability to compare entities in text using text-to-text generation objectives based on the data collected through (1). Specifically, they train an LM to learn to generate comparative questions, question-answer pairs, and summaries, while balancing this with a text infilling task over documents collected through (1).
3. A novel benchmark for this task, which was extracted from subsets of other benchmark datasets and websites.

This paper is mostly successful in demonstrating that their data collection approach can improve the performance of LMs on comparative reasoning. While I am struggling to contextualize the findings of this work with the advances in the most recent LLMs, I hope the authors will provide some clarity.

**Questions For The Authors:**

*Question A:* I didn’t quite understand the objectives presented in 3.3.1 and 3.3.2. Is the model learning to generate these questions and answers in an unsupervised manner through the given objective, or did you collect ground truth question data?

*Question B:* There isn’t much said about the ChatGPT evaluation in Table 3. What is the purpose of including ChatGPT in the evaluation? How exactly is it evaluated, e.g., did you use greedy decoding? Were some examples given in the prompt? What is the message of the ChatGPT results?

**Reasons To Accept:**

*Strength A.* The paper presents substantial resources for research in comparative reasoning, building a foundation for future research on a problem that is seemingly difficult for current SOTA language models.

*Strength B.* The proposed LM pre-training paradigms demonstrate sharp performance improvements on comparative reasoning tasks over vanilla counterparts (and sometimes even ChatGPT). Performance gains are especially apparent in lower-resource settings, which may be common as comparative reasoning annotation is expensive to collect and can be domain-specific. The evaluations used for comparative reasoning seem mostly sound.

*Strength C.* Analysis of pre-training objectives shown in Table 4 and Section 3.4 demonstrate the benefit brought by each objective, further strengthening the authors’ contribution.

**Reasons To Reject:**

*Weakness A.* While the evaluations show that the authors’ proposed pre-training paradigms help improve the performance of T5 and BART, it’s unclear how to contextualize results with the current paradigm of using larger language models, e.g., ChatGPT, in zero-shot and in-context learning settings to tackle language tasks. Based on the results in Tables 3-4, it seems that their proposed approach really sees big benefits when applied in a zero-shot setting on downstream comparative reasoning tasks. This finding may be relevant to this paradigm, but the authors don't give any details or discussion about their evaluation of ChatGPT in Table 3, which makes it difficult to judge whether it’s comparable to their zero-shot results.

**Reproducibility:**

4: Could mostly reproduce the results, but there may be some variation because of sample variance or minor variations in their interpretation of the protocol or method.

**Reviewer Confidence:**

3: Pretty sure, but there's a chance I missed something. Although I have a good feel for this area in general, I did not carefully check the paper's details, e.g., the math, experimental design, or novelty.

**Typos Grammar Style And Presentation Improvements:**

The abstract may be more clear if it starts with a sentence about what comparative reasoning is and why it’s important, e.g., “Comparative reasoning is the process of  … which can enable …”.

L023: do you have a citation for this?

In the results tables, it may help to explain in captions that R-L and R-2 refer to ROUGE metrics.

In some spots, e.g., Table 3 caption, you mention performance differences are “significant”. You may want to provide some statistical significance testing to justify this, as not all performance differences are huge.

---

> ### Author Rebuttal · Authors · 2023-08-29
>
> Thanks for acknowledging the value of our study! In terms of your concerns, here are our clarifications:
> - **Weakness A & Question B**: Since ChatGPT is a much larger model with much more pre-trained data, and the downstream datasets might have leaked to its training data, it is not a comparable baseline. We provide ChatGPT as a reference to the performance of one of the most advanced large language models.
> We prompt ChatGPT (gpt3.5-turbo) in zero-shot setting and the prompts are shown in the below table. We modify the prompts from the ones we use with T5/BART and empirically choose the ones that work effectively, otherwise ChatGPT will generate too lengthy responses. We appreciate your comment and will add experimental details to the appendix.
>
> Here's a table showing the prompts used with ChatGPT.
> | Task | Prompt |
> | ----- | ---- |
> | Comparative QA (HotpotQA, 2WikiQA) |  Paragraph: [D1, D2] \n Question: [Q] \n Only output the short answer. |
> | Comparative QG (HotpotQG, 2WikiQG) | Paragraph: [D1, D2] \n Given the paragraphs above, please output a factoid question that requires comparison of the two entities on some shared property. |
> |  Comparative Summ (CocoTrip) |  Reviews of Hotel 1: [D1] \n Reviews of Hotel 2: [D2] \n Given the reviews of the two hotels above, please output a summary of the common opinions about both hotels, which only contains subjective information that is commonly described in both sets of the reviews. The summary should be less than 30 words. |
> | Comparative Summ (Diffen) | Paragraph 1: [D1] \n Paragraph 2: [D2] \n Based on the paragraphs above, output a comparative summary of the two entities. The summary should be less than 100 words. |
>
> - **Question A**: *"Is the model learning to generate these questions and answers in an unsupervised manner through the given objective, or did you collect ground truth question data?"*  In Section 3.3.1 and 3.3.2, we pre-train the model with the data collected by our labor-free and scalable framework. The QA pairs are textualized from the entity comparison quintuples using pre-defined templates so it doesn't involve human-labeled ground truth. Detailed description about the textualization can be found in Section 3.2.3.
>
> Hope our response addresses your concerns! We appreciate your feedback and are open to further discussion.

---

### Official Review · Reviewer_5Nu8 · 2023-08-03

**Soundness:** 3

**Excitement:**

3: Ambivalent: It has merits (e.g., it reports state-of-the-art results, the idea is nice), but there are key weaknesses (e.g., it describes incremental work), and it can significantly benefit from another round of revision. However, I won't object to accepting it if my co-reviewers champion it.

**Missing References:**

These works represent the most influential ones I can think of for synthetic question generation. I would advise reading and following their citation trails as well:
- Unsupervised Multi-hop Question Answering by Question Generation by Pan et. al 2020
- Improving Unsupervised Question Answering via Summarization-Informed Question Generation by Lyu et. al 2021
- Synthetic QA Corpora Generation with Roundtrip Consistency by Alberti et. al 2019
- PAQ: 65 Million Probably-Asked Questions and What You Can Do With Them by Lewis et. al. 2021

**Paper Topic And Main Contributions:**

This paper proposes a new method for pre-training to improve on comparative question answering, question generation, and summarization. They do this by mining entity co-occurrences from various sources (Wikipedia, Gigaword, etc.) and use those mentions and the entities Wikipedia pages to automatically construct synthetic pre-training data.

They pre-train T5 and BART on this data and show improved performance over the base models with no pre-training.

**Reasons To Accept:**

- The paper is well written and the results clearly support the view that this approach improves over the baseline models
- The Diffen dataset looks particularly useful for future work on this topic
- They gather a large group of datasets to serve as a benchmark for comparative reasoning

**Reasons To Reject:**

- The framing of this paper suggests that comparative reasoning is a unique phenomena that merits specialized evaluation and pre-training. I think that could be a valid premise, but I’m not sure the evidence in this paper establishes that. For example, it seems to blur the lines between multi-hop QA and comparative reasoning - what is the difference? It seems like all multi-hop QA would be comparative.
- There are no comparisons to other works on synthetic QA generation - my guess is any other technique would perform similarly. My biggest concern is this lack of baselines, given the number of papers on the topic, I would hope to see at least one comparison (see references). It would also be important for the authors to cite some of these.
- The “pre-training” pipeline is very similar to some of the datasets evaluated on. For example, HotpotQA (and 2WikaQA) was constructed by having humans annotate questions in nearly the same manner. Thus, it is not very useful to know that using Hotpot-like training data would improve performance on Hotpot (the more interesting results on CoCoTrip or Diffen, I think). Plenty of papers have shown that more synthetic similar data is better (see references). The authors provide a great analysis of entity overlap in the appendix, but I think that’s a distinct issue from whether having more data is better. Without the improvements from 2Wiki and Hotpot the  methods gains are much more minor.
- The Diffen dataset is not discussed much and no examples are given. Unclear as to the quality of the dataset.
- The Diffen dataset and pre-training/mining pipeline is not mentioned to be open-sourced? Happy to remove this point if the authors are doing that

**Reproducibility:**

4: Could mostly reproduce the results, but there may be some variation because of sample variance or minor variations in their interpretation of the protocol or method.

**Reviewer Confidence:**

4: Quite sure. I tried to check the important points carefully. It's unlikely, though conceivable, that I missed something that should affect my ratings.

**Typos Grammar Style And Presentation Improvements:**

The data example in the appendix should be moved up (IMO) when the authors have more space - not an issue right now. It was very helpful!

Very minor: it could be interesting to make templates that are more diverse, perhaps ChatGPT generated templates. Could improve the results - but less important than having baselines.

---

> ### Author Rebuttal · Authors · 2023-08-29
>
> Thank you for acknowledging the value of our proposed collection pipeline of entity comparison data and our gathered downstream benchmark. We are also excited about comparative reasoning applications like the Diffen dataset.
>
> Response to reviewer's concerns:
> - *"For example, it seems to blur the lines between multi-hop QA and comparative reasoning - what is the difference? It seems like all multi-hop QA would be comparative."* First, we need to clarify that our research problem is not just multihop QA, but a wide range of applications involving comparative reasoning. Comparative reasoning is an important intelligent ability that is needed in the comparison multihop QA but also in many other tasks beyond QA. For example, we have included comparative QG and summarization in this study, but our collected quintuples for entity comparison can be easily adapted to other comparative reasoning tasks, which is valuable for future research. To further clarify, existing research on multi-hop QA mainly focused on retrieving the relevant document that contains the final answer. Differently, our work focuses particularly on the improvement of comparative reasoning ability of the LM itself over the retrieved relevant documents. Thus, our work is orthogonal to existing multihop QA work and our findings can be added to their work to yield further improvement.
> - *"There are no comparisons to other works on synthetic QA generation"* *" it is not very useful to know that using Hotpot-like training data would improve performance on Hotpot"* Thanks for pointing out these relevant papers in synthetic QA generation, whose findings of the effectiveness of QA templates well align with ours. However, our work is in a different line from theirs: Our main contribution is the philosophy and the novel pipeline of collecting quintuples that store entity comparison knowledge for comparative reasoning. We design multiple approaches to textualize the knowledge for LM pre-training for multiple downstreams. Table 4 shows all 3 pre-training tasks are effective, not just QA generation. Furthermore, our collected quintuples can be easily adapted to other comparative reasoning tasks by varying the textualization approaches. However, we highly appreciate the references you listed and we find L Pan 2020 is the most relevant. They use similar QA templates but only on 4 pre-defined properties, while we propose a novel pipeline to mine the entities comparison knowledge from news and Wikipedia, which covers a much wider range of properties. What could make a fair comparison is: their T5 based comparison-only model reaches 38.2/45.0 on EM/F1 while ours reaches 44.25/52.62 on EM/F1. Other papers (Alberti et al, Lyu et al, and Lewis et al) didn't work on multi-hop QA so we considered them as related work but not baseline methods. Again, these works are **orthogonal** to ours and they can be developed upon our approach. We will cite the papers and add above discussions in our next revised version.
> - *"The Diffen dataset is not discussed much and no examples are given"* Thanks for acknowledging the value of Diffen dataset. An example is given in Table 5. We will give more examples in the appendix in the next revised version.
> - *"The Diffen dataset and pre-training/mining pipeline is not mentioned to be open-sourced?"* We have attached the code in additional materials for review session. We'll release the code and dataset upon acceptance.
>
> Hope our response addresses your concerns! We appreciate your feedback and are open to further discussion.

---

### Official Review · Reviewer_EaYH · 2023-08-13

**Typos Grammar Style And Presentation Improvements:** 1) In Appendix B
**Soundness:** 4

**Excitement:**

3: Ambivalent: It has merits (e.g., it reports state-of-the-art results, the idea is nice), but there are key weaknesses (e.g., it describes incremental work), and it can significantly benefit from another round of revision. However, I won't object to accepting it if my co-reviewers champion it.

**Paper Topic And Main Contributions:**

This paper proposes an approach to collect data automatically for comparative reasoning using both structured and unstructured data. They use this data as pre-training for T5 and BART and show improvements across three areas 1) fine-tuning 2) few-shot 3) zero-shot.  With the largest gains coming from zero-shot and small gains coming from fine-tuning. For evaluation, they make another contribution by creating a benchmark of 3 tasks (QA, QA generation, and summarization) for comparative reasoning.

**Questions For The Authors:**

1) How do you do zero-shot and few-shot with T5-base? I am surprised it works at all since it's trained with a denoising objective. Moreover, the base model is pretty small comparatively to models conventionally used for zero-shot which are in the billions of parameters.
2) You have a ChatGPT baseline, but I do not see details on how it was done. For instance, was it zero-shot or few-shot?
3) Looking at Table 7, it seems the Q/A has redundant information. For instance Q2 implies Q1 and Q3 and Q5 are essentially the same. Do you think that affects the model quality? At the least it could cause slower training.
4) How much training data was possible? The pretraining set is fairly small (around 200k examples), was this the entirety of what could be obtained? When I think of LM pretraining, I think of scales at least in the orders of millions, this seems more like pre-fine-tuning contrary to the framing in the paper.
5) What about catastrophic forgetting? Do the pertained T5/BART models retain their abilities?

**Reasons To Accept:**

1) The paper makes contributions in three different areas related to comparative reasoning: 1) An approach to automatically create training data using Wikipedia and Wikidata for training large language models. 2) Experiments showing the utility of using the curated data and 3) An evaluation benchmark for comparative reasoning involving 3 tasks and 4 datasets.
2) The results show that pertaining on the their data creates models with stronger comparison reasoning abilities.
3) The paper is very thorough, I like the experiments on showing that few-shot performance scales with the amount of training and improvements aren not simply from entity matching are examples of the paper's completeness.

**Reasons To Reject:**

1) The pre-training in the paper is more like pre-fine-tuning as the dataset is fairly small ~200k examples and there are no experiments showing that the models retain their original abilities. Catastrophic forgetting could have happened for instance, making the models more like specially-tuned models for the tasks evaluated in the paper.
2) The tasks used for evaluation are the same as those used in pre-training (QA, generated QA, summaries). This paper feels a little like overfitting to a specific set of tasks via pretaining and my concern is that the models are no longer general and wouldn't have much utility.
2) Finetuning makes things very very close between the original models and the CMP ones which again argues that the models wouldn't have utility for other applications.
4) The summaries used in pertaining are generated from another pertained model. This is more like distillation than pretraining. One could use pretrained models for many of these tasks which would hamper the interestingness of the work.
5) No experiments are done with larger models, only base models are used. Do the results hold up with larger models? Few-shot and zero-shot are pretty weak for base models (and Id imagine even more so for models like T5 which are trained on a denoising objecting). The training data is pretty small, so experiments could likely be done on larger models to see if the results hold.
6) There are no comparisons or mentions of any related work or alternative methods. Also the fact that the authors had to create all of their evaluations concerns me if this problem is very niche or already covered by other tasks.

**Reproducibility:**

4: Could mostly reproduce the results, but there may be some variation because of sample variance or minor variations in their interpretation of the protocol or method.

**Reviewer Confidence:**

4: Quite sure. I tried to check the important points carefully. It's unlikely, though conceivable, that I missed something that should affect my ratings.

---

> ### Author Rebuttal · Authors · 2023-08-29
>
> Thanks for your thoughtful review!
>
> - **Weakness 1, 2**: According to the biggest concern on whether our pre-trained models have general utility for other tasks, we would like to clarify that our model is pre-trained for comparative reasoning generation, not for general language modeling. Therefore, it's unrealistic and unnecessary to require the model to have general utility to be fine-tuned for other language tasks. Please see reponse to Q4 for further clarification.
> - **Q1, Weakness 5**: *"How do you do zero-shot and few-shot with T5-base?"*  T5 is pre-trained with a multi-task mixture of unsupervised and supervised tasks, and all tasks are converted to a text-to-text format using different prompts. Therefore, T5 models have the ability to perform zero-shot and few-shot learning as long as carefully designed prompts are given, such as the ones we used in Table 2.  We adopt the prefixes during pre-training as well as downstream fine-tuning. Due to the cost of pre-training large LMs on large corpora, we leave it to future work.
> - **Q2**: *Experimental details of ChatGPT.* It's zero-shot with ChatGPT (gpt3.5-turbo). We modify the prompts from the ones we use with T5/BART and empirically choose the ones that work effectively, otherwise ChatGPT will generate too lengthy responses. Since ChatGPT is a much larger model with much more pre-trained data, and the downstream datasets might have leaked to its training data, it is not a comparable baseline. We provide ChatGPT as a reference to the performance of one of the most advanced large language models. We appreciate your comment and will add experimental details to the appendix in our revised version.
>
> Here's a table showing the prompts used with ChatGPT.
> | Task | Prompt |
> | ----- | ---- |
> | Comparative QA (HotpotQA, 2WikiQA) |  Paragraph: [D1, D2] \n Question: [Q] \n Only output the short answer. |
> | Comparative QG (HotpotQG, 2WikiQG) | Paragraph: [D1, D2] \n Given the paragraphs above, please output a factoid question that requires comparison of the two entities on some shared property. |
> |  Comparative Summ (CocoTrip) |  Reviews of Hotel 1: [D1] \n Reviews of Hotel 2: [D2] \n Given the reviews of the two hotels above, please output a summary of the common opinions about both hotels, which only contains subjective information that is commonly described in both sets of the reviews. The summary should be less than 30 words. |
> | Comparative Summ (Diffen) | Paragraph 1: [D1] \n Paragraph 2: [D2] \n Based on the paragraphs above, output a comparative summary of the two entities. The summary should be less than 100 words. |
>
> - **Q3**: *"For instance Q2 implies Q1 and Q3 and Q5 are essentially the same."* We adopted templates with diversified language to represent the same knowledge quintuple for better generalization, inspired by Flan-T5 (Chung et al, 2022).
> - **Q4**: *"The pretraining set is fairly small (around 200k examples), was this the entirety of what could be obtained?"* Yes. The amount of quintuples depends on the size and coverage of unstructured data. With Gigawords (V5), CC-News (Hugging Face version) and Wikipedia, the current amount (219K) of quintuples is the entire set that we collect. However, the data size does not limit our contribution. First, we propose a labor-free framework to collect knowledge for entity comparison instead of a perfectly pre-trained model. The data size can be easily expanded by applying our framework to other corpora. Second, Fig 2 shows the effect of pre-training data size. The performance grows linearly as the data increases on a logarithmic scale, as discussed in Section 4.4.
> - **Q4**: *"The pre-training in the paper is more like pre-fine-tuning"* We appreciate your careful consideration regarding the terminology "pre-training". We had the same concern and carefully surveyed it before using the term. To our knowledge, pre-training is training on a large dataset using self-supervised or unsupervised objectives, aiming to learn representations that can be later fine-tuned for downstream tasks. General language modeling pre-training enables the model to perform language related tasks, but pre-training doesn't have to be restricted to language modeling. Task-adaptive pre-training on a smaller but directly task-relevant corpus has been studied and widely accepted (see ''Don't Stop Pretraining: Adapt Language Models to Domains and Tasks'' by S Gururangan et al, 2020). We have discussed related work that inject task-related knowledge into LMs through pre-training in Section 2.2. On the contrary, pre-finetuning is massively multi-task learning using human labeled data. Thus, we believe that "pre-training" would better describe our proposed framework.
> - **Q5**: *"What about catastrophic forgetting? Do the pertained T5/BART models retain their abilities?"* Again, we want to emphasize that we aim to pre-train LMs for comparative reasoning, not for general language modeling, so the ability to be fine-tuned for other language tasks is not in our consideration. To avoid catastrophic forgetting, we employ text infilling as the denoising task that aligns with the model's original pre-training, as described in Section 3.3.4.
> - **Weekness 6**: *"There are no comparisons or mentions of any related work or alternative methods."* Comparative reasoning is an important type of intelligence that humans have and humans would like machines to have. It involves various NLP applications. We are the pilot study to enable pre-trained LM with such ability.
> - **Weekness 4**: *"The summaries used in pertaining are generated from another pertained model. This is more like distillation than pretraining."* For data collection, we use an off-the-shelf data-to-text model to generate synthetic summaries given two **tuples** from our collected quintuples. This does not involve complex comparative reasoning since the data-to-text model is already given the correct tuples. In contrast, we pre-train our LM to generate this summary given two **documents**. This requires much harder comparative reasoning ability because the LM needs to attend to both documents, identify the comparable properties of two entities and generate a fluent and meaningful summary about the entities' commonalities and differences. In short, the collection of synthetic summaries is different from the pre-training objective of comparable summarization, and thus this is not distillation.
>
> Hope our response addresses your concern! We appreciate your feedback and are open to further discussions.

---

### Meta-Review · Area_Chair_mG1v · 2023-09-15

**Recommendation:** 4

**Metareview:**

After several rounds of discussion with the authors, the reviews lean positive on this paper (both reviewers who left 3 "soundness" scores indicated they were really 3.5s in in their comments). The reviewers highlight the thoroughness of the the analysis and the usefulness of this work as a resource for others working in comparative reasoning. There are some concerns about comparative reasoning as a task, specifically whether it should come at the expense of other quality aspects and how it relates to other tasks and datasets (that could potentially make a stronger evaluation).

---

### Decision · Program_Chairs · 2023-10-07

**Decision:**

Accept-Main

**Comment:**

After several rounds of discussion with the authors, the reviews lean positive on this paper (both reviewers who left 3 "soundness" scores indicated they were really 3.5s in in their comments). The reviewers highlight the thoroughness of the the analysis and the usefulness of this work as a resource for others working in comparative reasoning. There are some concerns about comparative reasoning as a task, specifically whether it should come at the expense of other quality aspects and how it relates to other tasks and datasets (that could potentially make a stronger evaluation).